# Process evaluation protocol for the I-WOTCH study: an opioid tapering support programme for people with chronic non-malignant pain

Vivien P Nichols  ,[1] Charles Abraham,[2] Sam Eldabe,[3] Harbinder K Sandhu,[1] Martin Underwood,[1] Kate Seers,[4] on behalf of the I-WOTCH team

¹Warwick Clinical Trials Unit, Warwick Medical School, University of Warwick, Coventry, UK
²Faculty of Medicine, Dentistry and Health Sciences, University of Melbourne, Melbourne, UK
³Department of Pain Medicine, The James Cook University Hospital, Middlesbrough, UK
⁴Warwick Research in Nursing, Division of Health Sciences, Warwick Medical School, University of Warwick, Coventry, UK

**Correspondence to**
Vivien P Nichols;
v.p.nichols@warwick.ac.uk

## ABSTRACT

**Introduction** The Improving the Wellbeing of people with Opioid Treated CHronic Pain (I-WOTCH) randomised controlled trial uses a multicomponent self-management intervention to help people taper their opioid use. This approach is not widely used and its efficacy is unknown. A process evaluation alongside the trial will help to assess how the intervention was delivered, looking at the dose of intervention received and the fidelity of the delivery. We will explore how the intervention may have brought about change through the experiences of the participants receiving and the staff delivering the intervention and whether there were contextual factors involved.

**Methods and analysis** A mixed methods process evaluation will assess how the processes of the I-WOTCH intervention fared and whether these affected the outcomes. We will collect quantitative data, for example, group attendance analysed with statistical methods. Qualitative data, for example, from interviews and feedback forms will be analysed using framework analysis. We will use a 'following a thread' and a mixed methods matrix for the final integrated analysis.

**Ethics and dissemination** The I-WOTCH trial and process evaluation were granted full ethics approval by Yorkshire and The Humber—South Yorkshire Research Ethics Committee on 13 September 2016 (16/YH/0325). All data were collected in accordance with data protection guidelines. Participants provided written informed consent for the main trial, and all interviewees provided additional written informed consent. The results of the process evaluation will be published and presented at conferences.

**Trial registration number** ISRCTN49470934; Pre-results.

## INTRODUCTION

### The Improving the Well-being of People with Opioid Treated Chronic Pain (I-WOTCH) study

The I-WOTCH study is a randomised controlled trial (RCT) testing the effectiveness and cost effectiveness of a patient-centred, multicomponent self-management intervention targeting withdrawal of strong opioids among those living with chronic non-malignant pain. Primary outcomes are activities of daily living measured by the Patient-Reported

### Strengths and limitations of this study

► Little is known about whether a multicomponent self-management intervention can help people taper their opioid use.
► Process evaluation allowed for exploration into how a study was implemented, how processes fared and whether these were carried out as intended.
► Qualitative interviews gave insight into how people experienced the study both from those delivering and receiving the intervention.
► Using a mixed methods approach will enable us to explore lines of argument across the trial data.

Outcomes Measurement Information System Pain Interference Short Form 8A and opioid dose reduction measured as a morphine daily dose equivalent at 12 months' follow-up. Details of this are included in the I-WOTCH RCT protocol paper.[1] Trial participants are identified from general practice records, using electronic searches and approached by letter. They are randomised into the control group, who receive the 'My Opioid Manager' self-help guide and a relaxation compact disc or in addition to the intervention group who are invited to attend 3 days of group activities, two one-to-one sessions with a clinical facilitator (usually a nurse) after day 2 and up to two follow-up telephone calls between day 3 and the last one-to-one session. I-WOTCH is a multisite trial requiring standardisation of training and delivery. The intervention is complex and multicomponent, including educational and behavioural change components. Any changes in medication are discussed with participants. All prescriptions continue to be issued by the participants' general practitioners. Outcomes will be assessed at four time points: baseline and 4, 8 and 12 months. The I-WOTCH protocol paper gives greater detail about the study.[1]

## Preliminary work

We did a formative process evaluation as part of an intervention pilot study which found that the randomisation and the control arm seemed acceptable and the paperwork was not reported to be burdensome. There were no reported cases of resentful demoralisation or complaints about the randomisation process.

The delivery of groups and intervention attendance showed that group delivery was feasible, though numbers were lower than expected. Strategies were put in place to improve this in the main study. Once people attended day 1, attendance was good for the remaining group days and one-to-one sessions. Those who could not attend the first group session (most often due to work commitments or poor health) were offered a different future group. A member of the process evaluation team (VPN) observed one pilot group and reported good group engagement and facilitation of group content. Discussions were well received by the participants attending. Feedback from participants was positive about the course, the most useful aspects being the gaining of new knowledge about opioids and pain within a supportive environment. The participants also said that they found the components of the course, which helped them change their thoughts and attitudes to their pain useful. Things that they would change about the group sessions centred on practical considerations such as the comfort of seating and better sound equipment. This feasibility work helped us to develop our logic model and specific components of the main trial process evaluation, such as our interview topic guides and fidelity paperwork to assess adherence and competence.

## Process evaluation

This paper describes a process evaluation that is being conducted as an integral part of the I-WOTCH trial.

We are doing a mixed methods process evaluation based on UK Medical Research Council (MRC) guidance to better understand how the intervention works.[2] Key foci of evaluation, as described by Steckler and Linnan, are context, (contextual factors that may affect the implementation), fidelity (whether the intervention was delivered as designed), dose delivered (the amount of intervention delivered), dose received (the amount of intervention received by the participants) and reach (who the participants are and where they come from).[3] We will assess fidelity, ascertaining whether the trial processes were conducted as per protocol so minimising possible type III errors, for example, when the outcomes of a study do not take into account an inadequate implementation of an intervention.[4] We will identify any delivery that deviates from the original design because these may be important when interpreting the trial results. We will also investigate (1) how the contexts (eg, different sites) of implementation affected delivery, (2) how implementation of the intervention was managed

| The problem | Intervention Aims | Intervention | Theory and Guidance | Interim Targets | Desired Outcomes |
|---|---|---|---|---|---|
| People with chronic non-malignant pain are taking opioids, which have side effects and are not effective in the long term. | To test the effectiveness and cost effectiveness of a patient-centred multicomponent self-management intervention targeting withdrawal of strong opioids on activities of daily living for people living with chronic non-malignant pain | **Manualised Intervention Delivery**<br>**Core pain management topics:**<br>• Acute versus Chronic pain<br>• Acceptance<br>• Attention Control and distraction<br>• the pain cycle<br>• Posture and movement advice<br>• Relaxation techniques<br>• Stress busting for health action planning, problem solving, pacing, SMART goal setting<br>• identifying and overcoming barriers to change<br>• Mindfulness<br>• Anger, irritability and frustration<br>• Communication Skills<br><br>**Core opioid specific topics:**<br>• The rationale of prescribing in chronic pain<br>• Opioid induced tolerance and need for dose escalation<br>• Evidence of usefulness of opioids short and long term<br>• Side effects of opioids short term and long term<br>• Case studies of successful discontinued opioid therapy<br>• Opioid withdrawal symptoms<br>• Advantages of slow supervised tapering<br>• Symptom management during tapering<br>• Pain control after opioids | Theory of Planned Behaviour<br><br>Social Cognitive Theory<br><br>Information Motivation and Behavioural (IMB model) skills<br><br>Patient Centred Communication<br><br>Motivational Interviewing | **Staff Training**<br> To facilitate groups, deliver individual tapering consultations and telephone support in an inclusive and non-judgemental manner<br>**Individual participant changes:**<br>a Knowledge of: opioids, withdrawal effects, chronic pain<br>b Fostering change: self-validation, legitimising pain, normalising expectations<br>c Motivation to change by: Improved self-efficacy, effective tapering<br>d Skills:<br>• General Self-Regulation<br>*Psychological skills*<br>*Identify reasons for negative emotions (anger /frustration /irritable)*<br>*Identify problems and solutions, barriers to change*<br>*Recognise errors in thinking/automatic thoughts*<br>*Goal setting, goal review*<br>*Physical skills*<br>*Promote body awareness, posture*<br>*Reduce muscle tension*<br>*Body awareness and core strength*<br>*Relaxation-contract relax*<br>• Pain Self-Regulation<br>Understand that pain and mood are linked – when is pain bearable and when not bearable.<br>Understanding of pain cycle, unhelpful emotions and behaviours<br>Using mind to relieve pain does not mean pain in mind<br>Distraction whilst relaxed<br>Focus mind away from pain<br>Mindfulness for pain<br>Managing flare ups<br>Need for stretching<br>• Communication Skills<br>How to communicate with General Practitioners (GPs) and Health Care Professionals (HCPs)<br>Listening skills - Active and giving feedback in communication-reward for help. | **Primary outcomes:**<br>Patient-Reported Outcomes Measurement Information System (PROMIS) Pain Interference Short Form (8A)(PROMIS-PI-SF-8A)<br><br>Daily morphine equivalent opioid dose |

**Figure 1** Logic model.

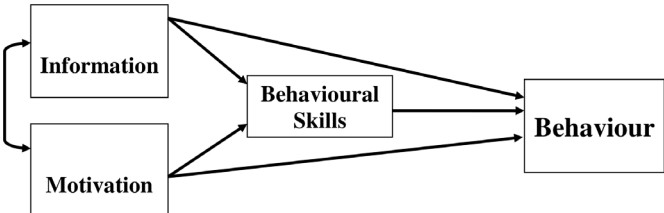

From J. D. Fisher and W. A. Fisher (1992). Changing AIDS risk behavior. Psychological Bulletin, 111, 455–74. Copyright by APA. Reprinted with permission.

**Figure 2** Information motivation behavioural skills model.

and (3) whether the hypothesised change mechanisms operated as expected. These data can inform replication, development and integration of interventions within routine practice so assisting researchers, commissioners and practitioners.

The process evaluation team (KS, VPN and CA), who has expertise in mixed methods approaches to complex health interventions, will work independently of the main trial team during the data collection phase to avoid contamination of trial processes. Findings from the process evaluation may provide insights that could enhance interpretation of the trial results.

### Aims
In summary, the aims of this process evaluation are to investigate
1. Experiences of the intervention, including enablers of, and barriers to, the intervention facilitating change among participants.
2. Intervention implementation, exploring the dose of the intervention delivered and received, and the fidelity of delivery.
3. Change mechanisms assessing whether the hypothesised change occurred.
4. Contextual issues that may affect the outcome or running of the study and/or intervention.

## METHODS
The aims of the I-WOTCH intervention are to improve participants' quality of life and to reduce their use of opioid drugs. We have developed a logic model specific to this intervention guided by intervention mapping principles[5] (see figure 1). We have also considered items from a checklist of key features of any group intervention that need to be reported to ensure replication enabling us to identify a series of characteristics to investigate.[6] The intervention includes educational, psychological and behavioural components designed to effect change. Thus, the change mechanisms can be conceptualised in terms of the information, motivation and behaviour skills model developed by Fisher and Fisher,[7] hypothesising that the intervention will (1) provide useful new information concerning the effects of opioids, (2) motivate participants to reduce their reliance on opioids and (3) provide them with new skills to facilitate non-opioid pain control (see figure 2).

We will use a mixed methods approach using quantitative data collected by the trial team, as well as qualitative data collection methods outlined in the following sections, which map into our aims.

### 1. Experiences of the intervention, including enablers of, and barriers to, the intervention facilitating change among the participants.
Qualitative data will include interviews with participants and those delivering the intervention. Participant feedback forms include qualitative and quantitative data from open and satisfaction questions, respectively (see table 1).

### Interviews
We will interview up to 20 intervention participants and 20 who were allocated to the control arm. These will be purposively sampled to ensure a range of age, gender, location and opioid reduction experience across the two trial arms. They will be interviewed after their final

| Table 1 | Interview and participant feedback data | |
| --- | --- | --- |
| **Key components** | **Source of data** | **Type of data** |
| Experiences of participants Interview topics including<br>► Responses to receiving the intervention or control.<br>► How they felt they were able to use it.<br>► How easy or difficult was it to use?<br>► Were some components more challenging than others?<br>► Specific barriers and enablers.<br>► Experience of being in a group (intervention only). | Participants | Interview recordings and transcripts |
| Participant feedback forms | Intervention participant forms | Feedback form questions |
| Experiences of delivering the intervention | Intervention delivery staff (clinical facilitator and lay person with chronic pain or allied health professional) | Interview recordings and transcripts |

**Table 2** Quantitative data on dose delivered and received

| Key components | Potential source of data | Type of data |
|---|---|---|
| Intervention groups | Trial data | Groups run, location and dates |
| Numbers attending each component of the three intervention days | Trial data | Attendance sheets per session |
| Uptake of the one-to-one sessions | Trial data / intervention staff | Intervention trial log<br>Staff interviews |
| Uptake of the telephone follow-up telephone calls. | Trial data / intervention staff | Intervention trial log<br>Staff interviews |

follow-up (at 12 months) to minimise possible effects of the interview on the trial findings. The interviews will be semistructured and held in a convenient local venue to the participant.

Participants will have agreed to be contacted about a possible interview on their initial trial consent form, and after receiving the 12-month follow-up questionnaires, our sample will be sent a patient information leaflet inviting them to take part in a face-to-face interview about their experiences of being part of the study. After a week, a researcher will contact them by phone to answer any questions and, if agreeable, book an appointment. A separate informed consent process will be completed at the beginning of each interview.

We will interview up to 20 trained intervention staff across different geographical areas. These include clinical facilitators (usually a nurse) whose role is to facilitate groups, see participants for their one-to-one appointments and give them telephone support as required. We will also interview the other group facilitators, either a lay person with experience of opioid use and tapering or an allied health practitioner with an interest in chronic pain conditions. Approach will be by an invitation letter with an information leaflet, and consent will be taken before the interview. Interviews will be semistructured using a topic guide and will take place after the interventions have been completed.

All interviews will be audio-recorded and transcribed verbatim when all identifiable data have been removed. They will then be checked for accuracy by the interview researcher. Audio recordings will be held in a digitally secure environment with restricted access.

We will analyse the interviews using both thematic analysis and framework analysis.[8 9] Transcripts will be analysed using the six steps outlined by Braun and Clarke.[9] After thorough familiarisation with the data through listening to all recordings, and reading and rereading the transcripts, five interviews will be analysed by coding themes related to the research questions. The emerging lower-level codes will then be grouped into higher level themes related to the research questions. All transcripts will then be coded using the hierarchical coding framework, paying attention to any new themes and deviant cases. We will review data related to each code and theme, check and recode, if necessary, and define themes. Throughout the analysis, the analysis team will make reflective analytic memos and hold regular discussion meetings. We will use NVivo qualitative data analysis software (QSR International Pty) to organise the data.

### Feedback forms
Feedback forms will be given to intervention participants after their last group or at their second one-to-one session. These forms are anonymous and will be sent back to the Warwick Clinical Trials Unit in a stamped addressed envelope to ensure anonymity. These forms contain quantitative satisfaction questions that will be analysed statistically (see analysis of data section) and open questions that will be analysed using thematic analysis (see online supplementary appendix 1).

### 2. Intervention implementation exploring dose of the intervention delivered and received, and the fidelity of delivery.
We will note the uptake and attendance of the different components of the intervention, to allow assessment of the intervention dose delivered and received (see table 2).

### Fidelity of intervention delivery
Fidelity will be assessed by rating facilitators' adherence to a detailed course manual and competency of delivery as taught in their training. All intervention sessions and

**Table 3** Fidelity of intervention delivery

| Key components | Source of data | Type of data |
|---|---|---|
| Assess fidelity of group sessions<br>▶ Adherence.<br>▶ Competence. | Audio recordings of all sessions | Adherence and competence ratings with researcher notes from a selection of sessions |
| One-to-one sessions<br>to understand the issues discussed | Audio recordings of all sessions | Adherence and competence ratings with researcher notes from a selection of first and second interviews |

**Table 4** Course programme with sessions identified for fidelity

| Day 1 |
| --- |
| Session 1: Introduction |
| Session 2: Pain information* |
| Session 3: Painkiller information and opioid education* |
| Session 4: Acceptance: John's story* |
| Session 5: Attention control and distraction |
| Session 6: Distraction activity—rose drawing |
| Session 7: Good days, bad days: when is pain bearable and when is it not?* |
| Session 8: The pain cycle of unhelpful emotions and behaviours* |
| Session 9: Posture |
| Session 10: Relaxation and breathing |
| Session 11: Summary of the day |
| **Day 2** |
| Session 12: Reflections from day 1 |
| Session 13: Stress busting—prioritising what's important, action planning, goal setting and pacing* |
| Session 14: Withdrawal symptoms, case studies (opioid education 2)* |
| Session 15: Distraction activity—origami |
| Session 16: Identifying and overcoming barriers to change, part 1—recognising unhelpful thinking* |
| Session 16: Identifying and overcoming barriers to change, part 2—reframing negatives to positives* |
| Session 17: Mindful attention control |
| Session 18: Balance and introduction to stretch |
| Session 19: Summary of the day |
| **Day 3** |
| Session 20: Reflections from day 2 and the previous week |
| Session 21: Anger, irritability and frustration* |
| Session 22: Relationships, part 1—getting the most from your healthcare team* |
| Session 22: Relationships, part 2—listening skills |
| Session 23: Managing setbacks and non-drug management techniques* |
| Session 24: Distraction activity—mindfulness colouring |
| Session 25: Stretching muscles that commonly get tight |
| Session 26: Mindfulness of thoughts and senses |
| Session 27: Summary of day 3 |
| Session 28: Summary of the course |

| Legend | *Educational and/or self-management regarding pain or opioid use | Practical, reflection or summarising sessions |
| --- | --- | --- |
| Day 1 | 2, 3, 4, 7 and 8 | 1, 5, 6, 9–11 |
| Day 2 | 13,14 and 16 | 12,15,17–19 |
| Day 3 | 21; 22, part 1; 23 | 20; 22, part 2; 24–28 |

one-to-one consultations will be audio recorded for the purpose of fidelity. This will be carried out by members of the process evaluation team (VPN and KS) listening to audio recordings of a sample of group and one-to-one sessions (see table 3).

The I-WOTCH main study protocol originally had a target of 468 participants (234 of whom will be allocated to the intervention) and anticipated running 24 groups.

To ensure a random sample of groups for the fidelity study, a statistician using a random number generator will identify three day 1 sessions, three day 2 sessions and three day 3 sessions from early, middle and late stages of the study. This will ensure we listen to approximately 10% of group sessions across the duration of the study. It was not possible to listen to all the sessions due to pragmatic reasons of time and cost. Through extensive discussions with the team who developed the intervention, we decided to prespecify those sessions that were considered by the team to be key to promoting behaviour change and contain either educational or discussion items. Other sessions that are more practical in nature (eg, origami for distraction or relaxation) will be difficult to assess from an audio recording as the aim was to promote distraction and discussion or to experience a relaxation technique. These will be checked to see if they took place as a minimum requirement of the intervention but will not be rated for facilitator adherence and competence (see table 4).

To assess the fidelity of the intervention, we will assess two aspects of intervention delivery: adherence to the intervention manual and competency of the facilitators. A member of the process evaluation team (VPN) will listen to the relevant recordings and score adherence and competence using a specially devised checklist based on components specified in the manual items and training protocol for intervention facilitators (see, eg, online supplementary appendices 2 and 3).

We will also rate one of the first or second one-to-one nurse consultations per group (n=24). We will double rate 10% of these sampled group and one-to-one sessions by a second member of the team (KS) to assess inter-rater

**Box 1    Motivation, expectation, self-efficacy and perceived intervention efficacy questions**

**Baseline motivation (baseline and follow-up).**
*I want to reduce my opioid use.*
(Not at all, by a little, by half, so I only use a little, so I use no opioids)

**Baseline expectation (baseline only).**
*I expect that, in 4 months' time, I will have reduced my opioid use.*
(Not at all, by a little, by half, so I only use a little, so I use no opioids)

**Baseline self-efficacy (baseline only).**
*I am confident I could reduce my opioid use a lot over 4 months.*
(Not at all confident, somewhat confident, fairly confident, strongly confident, completely confident)

**Perceived intervention efficacy (baseline and follow-up).**

**Baseline**
*I feel that involvement in this study can help me to reduce my opioid use.*
(Not at all, by a little, by half, so I only use a little, so I use no opioids)

**Follow-up**
*I feel that involvement in this study has helped me to reduce my opioid use.*
(Not at all, by a little, by half, so I only use a little, so I use no opioids)

reliability and to ensure rigour. Percentage scores will be given for adherence and competence per session, and the findings will be analysed using standard statistical methods (see the Mixed methods analysis section).

### 3. Change mechanisms assessing whether hypothesised change occurred.

We will administer self-report questions within the I-WOTCH RCT questionnaires (at baseline and 4, 8 and 12 months) to track possible change mechanisms. Specifically, we will assess participants' (1) motivation to reduce opioid use before and after the intervention, (2) expectations of success in opioid reduction, (3) confidence (or self-efficacy[10]) in relation to opioid reduction prior to receiving the intervention and (4) perceived intervention efficacy before and after participation (see box 1). These data will be analysed using standard statistical methods, including t-tests and analysis of covariance. The technical issues of the statistical analyses will be detailed in the overall trial statistical analysis plan.

### 4. Contextual issues that may affect the outcome or running of the study and/or intervention

Contextual factors may be found in the data collected previously may influence change and outcomes. We will explore this as the need arises from the data as it may be a 'thread to follow' (see next paragraph) or an integral part of a section of the analysis.

### Mixed methods analysis

Quantitative data will be analysed statistically to produce appropriate descriptive statistics, tables, charts or figures. Data from quantitative and qualitative findings will be integrated as outlined by O'Cathain *et al*.[11] We will use both 'following a thread', which involves selecting a question or component from one aspect of the findings and following across, and 'mixed methods matrix', where, for example, responses on quantitative scales can be compared with the interview transcript, and data on each case can be concisely stated and recorded on a matrix. For detailed explanation of following a thread, we refer the reader to O'Cathain.[11]

### Patient and public involvement

This process evaluation is part of the I-WOTCH study, which has patient and public involvement with regard to input into its design, as well as the ongoing running of the study, which is described more fully elsewhere.[1] Patient participant interviews are an integral part of this process evaluation. All trial participants will be notified of the study findings via a study newsletter, and a lay summary will be available on the study website.

### Trial status

The I-WOTCH study began recruitment in May 2017, and anticipated groups will be running into February 2019. Data collection will be completed around February 2020, and they expect the final report for the funders will be submitted mid-2020.

## ETHICS AND DISSEMINATION

We intend to publish the process evaluation findings in peer-reviewed journals, and details of the main trial ethics and dissemination are outlined in the main trial protocol.[1]

The I-WOTCH trial and process evaluation have been granted full ethics approval by Yorkshire and The Humber—South Yorkshire Research Ethics Committee on 13 September 2016 (16/YH/0325).

**Collaborators** I-WOTCH team: Sharisse Alleyne, University of Warwick. Shyam Balasubramanian, Department of Anaesthesia & Pain Medicine University Hospitals Coventry and Warwickshire NHS Trust. Lauren Betteley, University of Warwick. Katie Booth, University of Warwick. Dawn Carnes, Barts & The London Queen Mary's School of Medicine & Dentistry. Andrea Dompieri Furlan, Toronto Rehabilitation Institute, University Health Network, Canada. Kirstie Haywood, University of Warwick. Ranjit Lall, University of Warwick. Andrea Manca, The University of York. Dipesh Mistry, University of Warwick. Jennifer Noyes, The James Cook University Hospital. Anisur Rahman, University College London. Jane Shaw, The James Cook University Hospital. Nicole Kit Yee Tang, University of Warwick. Stephanie Taylor, Barts & The London Queen Mary's School of Medicine & Dentistry. Colin Tysall, University/User Teaching and Research Action Partnership, University of Warwick. Emma Withers, University of Warwick. Cynthia Paola Iglesis Urrutia, The University of York.

**Contributors** KS, MU, SE and HKS conceived the original study design. VPN, KS and CA have developed the study design and plan for data collection and analysis. All authors have provided critical revisions to the manuscript and approved the final manuscript.

**Funding** This project is funded by the National Institute for Health Research (NIHR), Health Technology Assessment (HTA) (project number 14/224/04). The views and opinions expressed therein are those of the authors and do not necessarily reflect those of the HTA, NIHR, NHS or the Department of Health.

**Competing interests** SE is an investigator on a number of National Institute for Health Research(NIHR) and industry-sponsored studies. He received travel expenses for speaking at conferences from the professional organisations. SE consults for Medtronic, Abbott, Boston Scientific and Mainstay Medical, none in relation to opioids. SE is chair of the BPS Science and Research Committee. SE is deputy Chair of the NIHR CRN Anaesthesia Pain and Perioperative Medicine National Specialty Group. SE's department has received fellowship funding from Medtronic as well as nurse funding from Abbott. HKS is director of Health Psychology Services Ltd, providing psychological services for a range of health-related conditions. MU was Chair of the NICE accreditation advisory committee until March 2017, for which he received a fee. He is chief investigator or coinvestigator on multiple previous and current research grants from the UK NIHR and Arthritis Research UK and is a coinvestigator on grants funded by the Australian NHMRC. He is an NIHR senior investigator. He has received travel expenses for speaking at conferences from the professional organisations hosting the conferences. He is a director and shareholder of Clinvivo Ltd, which provides electronic data collection for health services research. He is part of an academic partnership with Serco Ltd related to return to work initiatives. He is a coinvestigator on a study receiving support in kind from Orthospace Ltd. He is an editor of the NIHR journal series and a member of the NIHR Journal Editors Group, for which he receives a fee. KS received grant funding as PI and Col from NIHR for other projects and was on the NIHR HS&DR Funding Board until January 2018.

**Patient consent for publication** Not required.

**Provenance and peer review** Not commissioned; externally peer reviewed.

**ORCID iD**
Vivien P Nichols http://orcid.org/0000-0002-3372-1395

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
