## [Reviewer comments · BMJ Open]

ARTICLE DETAILS

TITLE (PROVISIONAL)	PROCESS EVALUATION PROTOCOL FOR THE I-WOTCH STUDY: AN OPIOID TAPERING SUPPORT PROGRAMME FOR PEOPLE WITH CHRONIC NON-MALIGNANT PAIN.
AUTHORS	Nichols, Vivien; Abraham, Charles; Eldabe, Sam; Sandhu, Harbinder; Underwood, Martin; Seers, Kate

VERSION 1 – REVIEW

REVIEWER	Edens Yale School of Medicine
REVIEW RETURNED	03-Feb-2019

GENERAL COMMENTS	Review of “Process Evaluation Protocol for the IWOTCH Study: An opioid tapering support programme for people with chronic non-malignant pain Summary: Building off of an initial pilot, the authors propose a process evaluation of a randomized controlled trial targeting self-management of opioid tapering. Importance: How to address a generation of patients who have been managed with long-term high dose opioid therapy is essential in addressing the current opioid crisis seen in developed nations. This implementation study will add to the knowledge regarding potentially effective interventions and future research directions. The aim and methods of the study appear sound and of interest to readership. I have only some relatively minor suggestions. Introduction: 1) This sentence wasn't clear to me. “Any changes in medication are discussed with, and if appropriate, any additional medications are prescribed by their GP.” What about, ‘All medication changes [I wondered if the authors meant, any ‘opioid’ medications? Or medications related to chronic pain?] are discussed with GP, who otherwise manages medical conditions per usual course.’ I actually wasn't entirely sure what the sentence was meaning to say—hence the suggestion to clarify. 2) Editing suggestion: “The deliver of groups and intervention attendance showed that group delivery was feasible, though numbers were lower than expected. Strategies were put in place to improve this in the main study.’ 3) Please clarify: “Once people attended day one attendance was good; and those who could not attend...” What about, ‘Once people attended Day 1, attendance on Day 2 was good.’? At least I'm assuming this is what the authors mean.
--

	4) In general, I would recommend changing from passive to active tense. For example, "Observation of one group reported good group engagement and facilitation of group content and discussions were well received by the participants attending." I found myself asking, 'Observation by whom? Who reported?' For clarity, I might suggest, "To ensure fidelity of the intervention, an outside observer attended one group intervention and reported it to have adequate group engagement and appropriate facilitation of the targeted content." 5) Check for inadvertently added/missing words. Methods: 1) In general, the methods appear sound. Qualitative interviews to determine key themes as well as fidelity assessments appear thorough and adequate. In thinking about change mechanisms, considering personal motivation, perceived confidence in reducing and perceived efficacy of any intervention appears intuitive. With regards to the 'expectations of success in opioid reduction', how is this different than 'confidence'? What about measuring participant beliefs about benefits or harms from opioid reduction? This is akin to motivation but might help elucidate change mechanisms if beliefs are changed pre- and post-intervention. 2) For consistency, please change past to present/future tense: "these include clinical facilitators, (usually a nurse) whose role is to..." 3) In paragraph under 'Change Mechanisms...', please review the numbering i-iv. 4) The readers might like a little more information about 'following a thread'. Perhaps several examples? Or being more explicit about the level of detail that is offered by the O'Cathain article. Suggestion: "For detailed explanation of 'following a thread', we refer the reader to O'Cathain which will be adequate to reproducing our design.' Or something like that.... Supplemental material 1) i-WOTCH Feedback form: For question #6: I would suggest not using the term facilitator, which might be an unfamiliar word to some. Perhaps, 'group leader'? Also, what about: "Overall the facilitators were..." versus 'Overall were the facilitators'?" I would suggest this change for #7 as well. References: 1) Appear adequate.
--	---

REVIEWER	Naomi Steenhof University Health Network & University of Toronto
REVIEW RETURNED	22-Feb-2019

GENERAL COMMENTS	The description of the preliminary work (formative process evaluation) was well described and convinced me of the feasibility of this research. I felt that the limitations could have been discussed in more detail (I could not find where in the manuscript they were discussed). I also would have appreciated more detail on how the team will decide the daily morphine equivalents if a participant is taking some proportion of their dose on an as needed basis. I had other questions with respect to whether the team will be collecting the indication for the participant's opioid use? This could have
--

	ramifications on the interpretation on your results (particularly decrease in opioid use). Will you be looking at the percentage decrease, overall decrease? or some other measure? Quite a bit of detail attached with respect to the program (Table 4). Overall this research is interesting, and as a practitioner in the field (who helps patients with opioid tapering) I would be very interested in seeing the results of this research.
--	---

REVIEWER	Mike Campbell University of Sheffield Uk
REVIEW RETURNED	19-Mar-2019

GENERAL COMMENTS	It is a good idea to do a process evaluation of a trial, and to set it up before the trial starts. This will be useful whether or not the trial shows and important outcome, if the intervention is little better than control, at least they will have a chance to know why!
---

REVIEWER	Peter Watson University of Cambridge UK
REVIEW RETURNED	07-May-2019

GENERAL COMMENTS	PROCESS EVALUATION PROTOCOL FOR THE IWOTCH STUDY: AN OPIOID TAPERING SUPPORT PROGRAMME FOR PEOPLE WITH CHRONIC NON-MALIGNANT PAIN bmjopen-2019-028998 I find the descriptions of statistical aspects of the study, the group sizes, randomised sampling of sessions, the analysis of quantitative data and other aspects vague and confusing. I would, therefore, like to see clarification in the text of all these points. Page 3, line 36 mentions the use of statistical methods. Is this referring to the qualitative analysis using NVivo or some other form of analysis such as quantitative analysis since quantitative data is to be collected by the trial team (line 18 on page 7)? If a quantitative analysis is anticipated please state what methods on what outcomes will be used. I further note a reference on page 9, lines 21-23 to statistical analysis of quantitative questions which again suggests the proposed use of an unspecified quantitative analysis approach. Page 5, line 19. In what way "was the randomisation and the control arm" acceptable as stated here? Pages 8 and 10. There appears to be a confusion on numbers proposed for the control and intervention groups. On page 8, lines 13-14 it states that there will be 20 participants in both groups. On page 10, lines 10-12 we now have 234 participants with half of these in the intervention group. Are these separate studies? What is the justification for the group sizes? Using 468 participants (line 10 on page 10) sounds very specific which makes me think some calculation (power or precision) may have been used to derive this figure. Page 10. I find the description of the randomisation confusing and am not clear how the randomisation would be carried out. What
---

	are the 24 groups (line 12 of page 10) which do not appear to have been defined elsewhere? I don't know how you are randomly sampling the day sessions in early, middle and late stages (lines 12-19 on page 10) of the study. What are these early, middle and late stages (line 17 of page 10) precisely? Do these correspond to the four time points of baseline, four, eight and twelve months mentioned on line 10 on page 5? Is more than one session sampled on a participant? Are you going to separately take random samples of participants within the specified numbers of sessions? 10% of group sessions seems a small number to sample (In addition I don't, I'm afraid, see how the random sampling of sessions equates to sampling 10% of group sessions). I think this random sampling of sessions applies only to the intervention but could you not consider also sampling sessions in the control group? In truth I don't understand from the text why they are sampling the sessions. Is it to assess validity in some way so that they can show that the sessions are working in the way they should? It is not proposed to assess (line 28 on page 10) rating facilitator adherence or competence but I assume the people leading the sessions such as the origami one (line 26 on page 10) are capable of leading a session. Origami is quite a complex skill and you need a teacher who can understand and demonstrate the folding techniques. I also notice from Table 3 (pages 9-10) that adherence and competence ratings are to be given for other activities but it does not say here how these ratings will be scored and who the raters will be. I assume some of the study co-authors will be the raters? If so, you could mention their initials e.g. will be rated by one of us who is a trained psychologist team members (insert initials of co-author doing he rating). I am rather concerned that one of the checking criteria (line 26 on page 10) is simply checking if a session takes place. I would expect in a study that at the very least the sessions which are proposed to form part of a study would actually occur. How do we know that other unsampled sessions are occurring? Is the holding of activity sessions feasible in terms of getting suitably qualified teachers and in activity sessions taking place? The English needs attention in places e.g. Page 10 lines 12-19 where an insertion of an 'a' and a full stop are both needed and page 11, line 12 and elsewhere uses the word 'fidelity' where I suspect they may mean validity. I wasn't clear what the (1) on page 5, line 12, (2) at page 5, line 57, (3) at page 6, line 3 and (4) at page 6, line 10 are referring to. I suspect it may be the aims of the project which are summarised on page 6, lines 33-47 but it is not clear.
--	--

VERSION 1 – AUTHOR RESPONSE

Reviewer comments	Author response
Reviewer: 1	

Importance: How to address a generation of patients who have been managed with long-term high dose opioid therapy is essential in addressing the current opioid crisis seen in developed nations. This implementation study will add to the knowledge regarding potentially effective interventions and future research directions. The aim and methods of the study appear sound and of interest to readership. I have only some relatively minor suggestions. Introduction: 1) This sentence wasn't clear to me. "Any changes in medication are discussed with, and if appropriate, any additional medications are prescribed by their GP." What about, 'All medication changes [I wondered if the authors meant, any 'opioid' medications? Or medications related to chronic pain?] are discussed with GP, who otherwise manages medical conditions per usual course.' I actually wasn't entirely sure what the sentence was meaning to say—hence the suggestion to clarify.	Thank you for your comments. We have clarified the text to show that the GPs are responsible for all changes in prescribed medication. Pg 5 1st paragraph
2) Editing suggestion: "The delivery of groups and intervention attendance showed that group delivery was feasible, though numbers were lower than expected. Strategies were put in place to improve this in the main study.'	We agree and have amended the text. Pg 5 3rd paragraph
3) Please clarify: "Once people attended day one attendance was good; and those who could not attend..." What about, 'Once people attended Day 1, attendance on Day 2 was good.'? At least I'm assuming this is what the authors mean.	We have altered the text for clarity. Pg 5 paragraph 3
4) In general, I would recommend changing from passive to active tense. For example, "Observation of one group reported good group engagement and facilitation of group content and discussions were well received by the participants attending." I found myself asking, 'Observation by whom? Who reported?' For clarity, I might suggest, "To ensure fidelity of the intervention, an outside observer attended one group intervention and reported it to have adequate group engagement and appropriate facilitation of the targeted content."	We have amended the text. Pg 5 paragraph 3
5) Check for inadvertently added/missing words.	We have checked the text.
Methods: 1) In general, the methods appear sound. Qualitative interviews to determine key themes as well as fidelity assessments appear thorough and adequate. In thinking about change mechanisms, considering personal motivation, perceived confidence in reducing and perceived efficacy of any intervention appears intuitive. With regards to the	Thank you for confirming the methods appear sound. We have thought about this carefully, and decided to keep our current approach here, as it is derived from the well established Borkovec & Nau approach. Borkovec TD, Nau SD. Credibility of analogue therapy

'expectations of success in opioid reduction', how is this different than 'confidence'? What about measuring participant beliefs about benefits or harms from opioid reduction? This is akin to motivation but might help elucidate change mechanisms if beliefs are changed pre- and post-intervention.	rationales. Journal of Behavior Therapy and Experimental Psychiatry 1972;3:257–260. However, these points can be explored within the qualitative interviews.
2) For consistency, please change past to present/future tense: “these include clinical facilitators, (usually a nurse) whose role is to...”	We agree and have amended. Pg 8 last paragraph
3) In paragraph under 'Change Mechanisms...', please review the numbering i-iv.	We have reviewed the numbering. Pg 12 1st paragraph
4) The readers might like a little more information about 'following a thread'. Perhaps several examples? Or being more explicit about the level of detail that is offered by the O’Cathain article. Suggestion: “For detailed explanation of ‘following a thread’, we refer the reader to O’Cathain which will be adequate to reproducing our design.’ Or something like that....	This is helpful, thank-you. We will revise this to “For detailed explanation of ‘following a thread’, we refer the reader to O’Cathain Pg 13 2nd paragraph
Supplemental material 1) i-WOTCH Feedback form: For question #6: I would suggest not using the term facilitator, which might be an unfamiliar word to some. Perhaps, 'group leader'? Also, what about: “Overall the facilitators were...” versus ‘Overall were the facilitators’”? I would suggest this change for #7 as well.	Thank you. We would prefer to keep the term facilitator, as we feel it more accurately reflects the group processes in play, rather than just being led.
References: 1) Appear adequate.	Thank you
Reviewer: 2 1) The description of the preliminary work (formative process evaluation) was well described and convinced me of the feasibility of this research. I felt that the limitations could have been discussed in more detail (I could not find here in the manuscript they were discussed.	Thank-you. We agree it is important to address limitations, and will do this in detail in the main paper
2 I also would have appreciated more detail on how the team will decide the daily morphine equivalents if a participant is taking some proportion of their dose on an as needed basis. I had other questions with respect to whether the team will be collecting the indication for the participant's opioid use? This could have ramifications on the interpretation on your results (particularly decrease in opioid use). Will you be looking at the percentage decrease, overall decrease? or some other measure?	We note the reviewer’s interest in how we are going to report the outcome of opioid use. This is not germane to the conduct of the process evaluation and is addressed in the protocol paper which we have referenced here. The full details of this will be in our statistical analysis plan. We also anticipate publishing a separate paper on how to calculate opioid equivalence.
Quite a bit of detail attached with respect to the program (Table 4). Overall this research is interesting, and as a practitioner in the field (who	Thank you for your comments. We are also very keen to disseminate our findings widely.

helps patients with opioid tapering) I would be very interested in seeing the results of this research.	
Reviewer: 3 It is a good idea to do a process evaluation of a trial, and to set it up before the trial starts. This will be useful whether or not the trial shows an important outcome, if the intervention is little better than control, at least they will have a chance to know why!	Thank you for your comments
Reviewer: 4 1 I find the descriptions of statistical aspects of the study, the group sizes, randomised sampling of sessions, the analysis of quantitative data and other aspects vague and confusing. I would, therefore, like to see clarification in the text of all these points.	Thank you for your comments we have clarified these points below.
2) Page 3, line 36 mentions the use of statistical methods. Is this referring to the qualitative analysis using NVivo or some other form of analysis such as quantitative analysis since quantitative data is to be collected by the trial team (line 18 on page 7)? If a quantitative analysis is anticipated please state what methods on what outcomes will be used. I further note a reference on page 9, lines 21-23 to statistical analysis of quantitative questions which again suggests the proposed use of an unspecified quantitative analysis approach.	We have added text for clarity. Pg 9 paragraph 4, signposting to the mixed methods analysis Pg13 paragraph 2
3) Page 5, line 19. In what way "was the randomisation and the control arm" acceptable as stated here?	We have added some explanatory text.Pg 5 paragraph 2
4)Pages 8 and 10. There appears to be a confusion on numbers proposed for the control and intervention groups. On page 8, lines 13-14 it states that there will be 20 participants in both groups. On page 10, lines 10-12 we now have 234 participants with half of these in the intervention group. Are these separate studies? What is the justification for the group sizes? Using 468 participants (line 10 on page 10) sounds very specific which makes me think some calculation (power or precision) may have been used to derive this figure.	We have added text to clarify on pg 8 and 10. The 20 participants from both arms of the main study are for the interviews within the RCT which had a sample size of 468
5) Page 10. I find the description of the randomisation confusing and am not clear how the randomisation would be carried out. What are the 24 groups (line 12 of page 10) which do not appear to have been defined elsewhere? I don't know how you are randomly sampling the day sessions in early, middle and late stages (lines 12-19 on page 10) of the study. What are these early, middle and late stages (line 17 of page 10) precisely? Do these correspond to the four time points of baseline, four, eight and twelve months mentioned on line 10 on page 5?	We have added text to clarify. Pg 10 end paragraph. The early middle and late phases are during the running of the RCT group interventions.

6) Is more than one session sampled on a participant? Are you going to separately take random samples of participants within the specified numbers of sessions? 10% of group sessions seems a small number to sample (In addition I don't, I'm afraid, see how the random sampling of sessions equates to sampling 10% of group sessions). I think this random sampling of sessions applies only to the intervention but could you not consider also sampling sessions in the control group?	Thank you for these comments we have identified an error in the numbers of the days and we have amended this. It should be three Day 1, 2 and 3 giving a total of nine . We have added text to clarify how and why we have sampled group intervention sessions. Pg 10 end paragraph. The control arm receive a My Opioid Manager booklet and relaxation CD only.
7) In truth I don't understand from the text why they are sampling the sessions. Is it to assess validity in some way so that they can show that the sessions are working in the way they should?	We have added text to clarify that the fidelity is being assessed from a random sample of group sessions. Pg 10 end paragraph
8) It is not proposed to assess (line 28 on page 10) rating facilitator adherence or competence but I assume the people leading the sessions such as the origami one (line 26 on page 10) are capable of leading a session. Origami is quite a complex skill and you need a teacher who can understand and demonstrate the folding techniques.	We have added text to clarify. Facilitators will be following a detailed manual and have been given training in how to deliver all sessions. Page 10 last paragraph The origami consists of a basic figure which has a step by step guide. The aim of the task is to promote distraction and discussion and often the facilitators will get involved too. During the facilitator training session, facilitators are given all of the material, they can then go away and practice before the course of they so wish to.
9) I also notice from Table 3 (pages 9-10) that adherence and competence ratings are to be given for other activities but it does not say here how these ratings will be scored and who the raters will be. I assume some of the study co-authors will be the raters? If so, you could mention their initials e.g. will be rated by one of us who is a trained psychologist team members (insert initials of co-author doing he rating).	Thank you for this comment. We have explained the scoring process in pg 11 end paragraph. We have added the initials of the members of the team who will be doing this for clarification and also the team's expertise. pg 6 paragraph 2
10) I am rather concerned that one of the checking criteria (line 26 on page 10) is simply checking if a session takes place. I would expect in a study that at the very least the sessions which are proposed to form part of a study would actually occur. How do we know that other unsampled sessions are occurring? Is the holding of activity sessions feasible in terms of getting suitably qualified teachers and in activity sessions taking place?	Thank you for your comment. We have clarified our reasons in the text. In the fidelity section on page 10. For the purposes of the fidelity study we cannot listen to all the group sessions due to reasons of time and cost. We decided that listening to 10% would give us an idea of how these groups were being run. Other data such as attendance sheets and facilitator interviews will give us further information about the running of the groups and whether sessions might have been missed. As mentioned in comment 8. The practical sessions

	have been taught to the facilitators in training and supplemented by a detailed manual with the resources required.
11) The English needs attention in places e.g. Page 10 lines 12-19 where an insertion of an 'a' and a full stop are both needed and page 11, line 12 and elsewhere uses the word 'fidelity' where I suspect they may mean validity. I wasn't clear what the (1) on page 5, line 12, (2) at page 5, line 57, (3) at page 6, line 3 and (4) at page 6, line 10 are referring to. I suspect it may be the aims of the project which are summarised on page 6, lines 33-47 but it is not clear.	We have checked for grammatical errors and amended these. Fidelity is defined in the introduction. The numbers mentioned are references in the BMJ modified Vancouver style
FORMATTING AMENDMENTS (if any) Required amendments will be listed here; please include these changes in your revised version: 1. Please re-upload your Supplementary files in PDF format.	This has been carried out
2. Please include Figure legends at the end of your main manuscript.	This has been carried out
3. The in text citation for "Table 1" is missing in your main text of your main document file. Please amend accordingly.	This has been added
4. Patient and Public Involvement: We have implemented an additional requirement to all articles to include 'Patient and Public Involvement' statement within the main text of your main document. Please refer below for more information regarding this new instruction: Authors must include a statement in the methods section of the manuscript under the sub-heading 'Patient and Public Involvement'. This should provide a brief response to the following questions: How was the development of the research question and outcome measures informed by patients' priorities, experience, and preferences? How did you involve patients in the design of this study? Were patients involved in the recruitment to and conduct of the study? How will the results be disseminated to study participants? For randomised controlled trials, was the burden of the intervention assessed by patients themselves? Patient advisers should also be thanked in the contributorship statement/acknowledgements. If patients and or public were not involved please state this.	Thank you for this comment we have added a section to explain that the I-WOTCH study design and ongoing running of the study has had PPI involvement. We have included patient input and dissemination of findings. Pg 13

--	--

VERSION 2 – REVIEW

REVIEWER	Peter Watson University of Cambridge UK
REVIEW RETURNED	02-Aug-2019

GENERAL COMMENTS	PROCESS EVALUATION PROTOCOL FOR THE IWOTCH STUDY: AN OPIOID TAPERING SUPPORT PROGRAMME FOR PEOPLE WITH CHRONIC NON-MALIGNANT PAIN bmjopen-2019-028998.R1 The paper is clearer however I have a few remaining queries as below. Page 8, line 31. I don't know what 'purposively sampling' is. Is this a form of minimisation to try and balance for age, gender and other possible confounders between the control and intervention groups? You could test to see if any of these confounders differ between the groups and, if so, adjust for these entering them as predictors in the ANCOVA as mentioned on page 12, line 22. Page 8, line 34 and Page 12, lines 10-12. I don't see any mention of handling dropout in the ANCOVAs between baseline and follow ups over the following 12 months. I wondered whether given you are performing hypotheses (Page 12, lines 10-22) looking at the effect of an intervention on the amount of change over time in motivation, expectations and confidence in opioid reduction in this clinical trial whether you had considered powering the study to obtain group sizes for the intervention and control groups assessing such hypotheses.
---

VERSION 2 – AUTHOR RESPONSE

Reviewers' comments	Response
Editorial request: - As previously suggested by Reviewer 1, please add a comma for clarity in this sentence: "Once people attended day one attendance was good..."  "Once people attended day one, attendance was good..." Reviewer 4 Comments to Author:	We have now added this.

The paper is clearer however I have a few remaining queries as below.

Page 8, line 31. I don't know what 'purposively sampling' is. Is this a form of minimisation to try and balance for age, gender and other possible confounders between the control and intervention groups? You could test to see if any of these confounders differ between the groups and, if so, adjust for these entering them as predictors in the ANCOVA as mentioned on page 12, line 22.

Page 8, line 34 and Page 12, lines 10-12. I don't see any mention of handling dropout in the ANCOVAs between baseline and follow ups over the following 12 months.

I wondered whether given you are performing hypotheses (Page 12, lines 10-22) looking at the effect of an intervention on the amount of change over time in motivation, expectations and confidence in opioid reduction in this clinical trial whether you had considered powering the study to obtain group sizes for the intervention and control groups assessing such hypotheses.

Thank you for your additional comments.

Purposive sampling is a qualitative term to ensure we invite participants with a range of different attributes such as age gender and opioid usage. We will be using qualitative analysis for the qualitative data from the process evaluation.

Page 8 line 34. The analysis of these interviews will be qualitative not quantitative and so no statistical description is needed.

Page 12 We have added to the text for clarity. 'The technical issues of the statistical analyses will be detailed in the overall trial statistical analysis plan.'

We will be doing between exploratory group analyses on changes in expectations and confidence between the two groups. These will however be exploratory rather than hypothesis testing. Details of the analytical approach will be covered in the overall statistical analysis plan for the trial.